# Curriculum as Selective Data Acquisition: Toward Reliable Generalization in Goal-Conditioned RL

## Abstract

We study curriculum learning in goal-conditioned reinforcement learning (GCRL) through the lens of data selection. Instead of sampling all goals uniformly, we bias sampling toward underachieved goals, thereby shifting the state–goal distribution seen by the agent. Using universal value function approximators (UVFAs) with potential-based reward shaping in GridWorld, we compare uniform and curriculum-guided training. Our results show that curricula alter goal coverage, reduce approximation error, and improve success on difficult edge goals. These findings highlight curriculum learning as a principled mechanism for selective data acquisition, suggesting a pathway toward more persistent and open-ended agents.

## 1 Introduction

Goal-conditioned reinforcement learning (GCRL) provides a flexible paradigm for training agents to solve multiple tasks within the same environment by conditioning policies or value functions on a goal state (Schaul et al., 2015). A persistent challenge in this setting is that many goals are difficult to reach under uniform sampling, leading to extremely sparse rewards and poor signal for function approximation (Andrychowicz et al., 2017). This challenge is magnified in open-ended learning (OEL), where agents must continually acquire and refine skills over an unbounded set of goals or tasks (Clune, 2019). Our own motivation for this paper stems directly from recent work by Hughes et al. (2024), who highlight the need for algorithmic paths toward persistent, open-ended learning. We view curriculum learning as one such path, offering a tractable starting point for shaping state–goal distributions.

Curriculum learning has been widely explored as a remedy for sparsity and exploration issues, typically by sequencing goals from easy to hard (Bengio et al., 2009; Florensa et al., 2017; Portelas et al., 2020). Prior work has developed handcrafted curricula (Bengio et al., 2009; Florensa et al., 2017), teacher–student frameworks (Matiisen et al., 2019; Narvekar et al., 2020), and automated goal-generation strategies (Held et al., 2018; Portelas et al., 2020). While these approaches differ in implementation, they share a core intuition: curricula act as a mechanism to ensure agents remain within their "zone of proximal development," preventing stagnation on trivial tasks and collapse on impossible ones (Matiisen et al., 2019).

Despite this progress, much of the literature treats curriculum as an exploration heuristic or as a way to overcome reward sparsity. Far less attention has been paid to its effect on the *distribution of training data* itself. In particular, curricula can be seen as a form of selective data acquisition, biasing the state–goal visitation distribution toward goals that are currently underachieved. This reframing highlights a structural rather than incidental role for curricula: by reshaping the data distribution, they change the inductive biases of the learned function approximator. By focusing on how curricula reshape state–goal distributions, we explicitly link curriculum design in GCRL to the broader questions of persistence and adaptability central to OEL (Hughes et al., 2024).

In this work, we investigate this perspective empirically using Universal Value Function Approximators (UVFAs; (Schaul et al., 2015)) trained in GridWorld. We compare uniform goal sampling to curriculum-biased sampling, analyzing how distributional shifts in data affect function approximation and downstream policy success. We show that curricula concentrate data in informative

regions of the state–goal space, reduce approximation error on a shared evaluation set, and improve policy success particularly on harder-to-reach goals. These findings suggest that curriculum learning should be understood not only as an exploration strategy, but also as a structural mechanism for guiding data acquisition—one that provides a concrete entry point into the larger challenge of scaling toward lifelong and open-ended learning.

## 2 METHODS

### 2.1 ENVIRONMENT SETUP

We use a deterministic GridWorld navigation environment where an agent must reach a goal location specified at the start of each episode. Each state is defined by the agent's current position, and each task is defined by a desired goal cell. Episodes terminate either upon reaching the goal or when a maximum horizon H is reached. This setting provides full observability, yet exposes the challenges of goal-conditioned reinforcement learning: large goal spaces, varying difficulty across cells (interior vs. edge), and sparse terminal rewards.

### 2.2 UNIVERSAL VALUE FUNCTION APPROXIMATORS (UVFAS)

We employ Universal Value Function Approximators (UVFAs; Schaul et al., 2015), which generalize value estimation across states and goals.

- Input: concatenation of agent state (x, y)    and   $(g_x, g_y)$

- Architecture: a multilayer perceptron (MLP) with ReLU activations and hidden dimension 64.

- Output: scalar estimate of the value function $V(s, g)$

- Training objective: mean squared error regression against pseudo-reward targets (see below).

This formulation allows us to assess not only policy performance but also how curricula affect function approximation quality across the entire state–goal space.

### 2.3 POTENTIAL-BASED REWARD SHAPING (PBRS)

To provide dense learning signals, we adopt Potential-Based Reward Shaping (PBRS; Ng et al., 1999). The formula is defined as follows:

Define a potential $\phi(s, g) = -d(s, g)$ where $d$ is the Manhattan distance between state and goal

The shaped reward is defined as:

$$r_t = \lambda[\gamma\phi(s_t + 1, g) - \phi(s_t, g)] - c$$

where discount $\gamma = .99$, shaping coefficient $\lambda = .5$ and step cost $c = .01$. A terminal bonus of $+1$ is added on successful episodes.

Targets are constructed as discounted returns-to-go under this shaped reward. For evaluation, we negate returns so that greedy action selection corresponds to $\arg\max$ over predicted values.

### 2.4 CURRICULUM DESIGN

We compare two data acquisition strategies:

1. Uniform (NoCurr): goals are sampled uniformly from all valid grid cells

2. Edge-Weighted Curriculum (Curr): sampling distribution is biased toward harder-to-reach goals, defined as those on the grid periphery. Empirically, edge cells are less frequently reached under uniform sampling, leading to underrepresentation in training data

In all cases, we collect fixed-size datasets per seed and train UVFAs with identical architectures, isolating the effect of curriculum-induced distributional shifts.

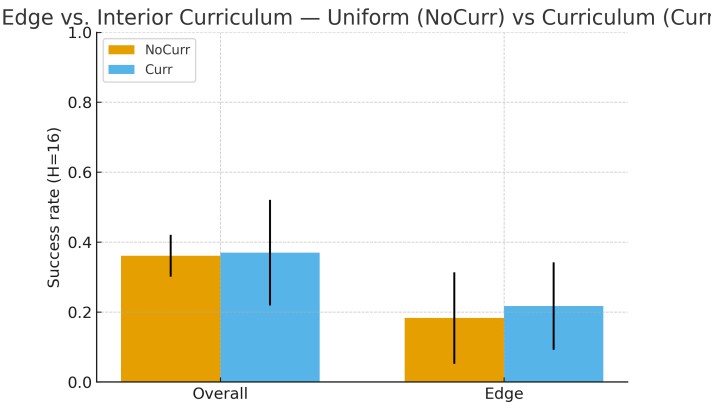

Figure 1: Edge vs. Interior Curriculum. Success rates at horizon $H = 16$ for agents trained with uniform sampling (NoCurr) versus edge-biased curriculum (Curr). Results are averaged across three seeds; bars show mean $\pm$ one standard deviation. Curriculum improves performance on harder edge goals while maintaining comparable performance overall.

## 2.5 Training Protocol

Data collection: For each seed, we roll out 1000 episodes with greedy action selection under PBRS shaping. Each trajectory is stored as a JSONL file and converted into a PBRS dataset (*.npz).

UVFA training: Models are trained for 50 epochs using Adam with learning rate $10^-3$ and batch size 256. Each run is repeated across three seeds for robustness.

Evaluation: Trained UVFAs are evaluated zero-shot on held-out goals with varying horizons ($H \in \{30, 20, 16, 12\}$).Success is measured as the fraction of goals achieved within horizon $H$, reported separately for interior and edge subsets.

## 3 Results

### 3.1 Baseline: Uniform vs. Curriculum Sampling

We first compare universal value function approximators (UVFAs) trained on uniformly sampled goals (NoCurr) with those trained using a manual curriculum that upweights edge goals (Curr). All agents were trained on $N = 1000$ episodes per seed (three seeds, max steps $= 30$). Evaluation was performed using greedy policies at varying horizons $H \in \{30, 20, 16, 12, 10\}$.

**Success rates** Across seeds, the curriculum models showed modest but consistent improvements on harder edge goals, with comparable overall performance. At $H = 16$, uniform (NoCurr) achieved $0.361 \pm 0.060$ overall and $0.183 \pm 0.131$ on edge goals, whereas curriculum (Curr) achieved $0.370 \pm 0.151$ overall and $0.217 \pm 0.125$ on edge goals (Fig. 2). While not universally stronger in aggregate, the curriculum condition tended to improve performance on the harder subset, consistent with the idea that selective sampling reshapes the state–goal distribution.

**Distributional shifts.** We confirm that edge-biased curricula shift the training distribution (Fig. 2), with increased density of trajectories targeting harder edge goals. These shifts translate into modest but measurable improvements in function approximation and policy success in these regions, supporting our hypothesis that curriculum should be viewed as selective data acquisition. While gains are not uniform across all goals, the evidence suggests that even simple hand-crafted curricula can systematically bias training data toward underachieved subsets and thereby improve performance where uniform sampling struggles.

To test the effect of curriculum design choices, we compared two variants against uniform sampling (NoCurr). The *baseline curriculum* biased sampling toward edge goals with a fixed proportion,

Figure 2: Edge vs. Interior Curriculum. Training distributions and success rates at horizon $H = 16$ for agents trained with uniform sampling (NoCurr) versus edge-biased curriculum (Curr). Results are averaged across three seeds; bars show mean $\pm$ one standard deviation. Curriculum biases data toward harder edge goals, yielding modest improvements in those regions while maintaining comparable performance overall.

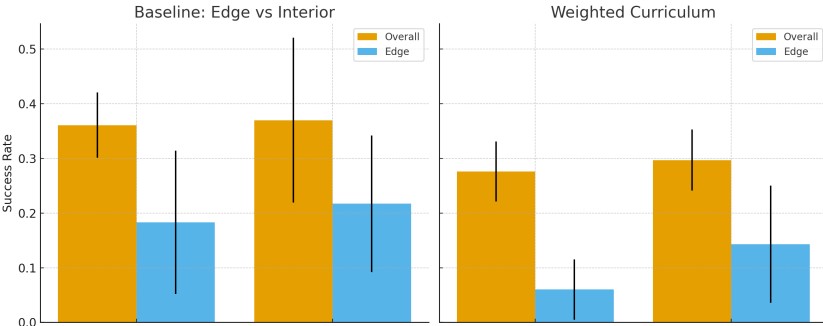

Figure 3: Curriculum variants. Success rates at horizon $H = 16$ for agents trained with uniform sampling (NoCurr), baseline curriculum (Curr), and weighted curriculum (Curr-W). Results are averaged across three seeds; bars show mean $\pm$ one standard deviation. Both curricula improve edge-goal success, with weighting amplifying the effect.

while the *weighted curriculum* further increased edge sampling to match their empirical difficulty under NoCurr. This weighting scheme was intended to strengthen the effect of curriculum as selective data acquisition, especially on harder goals.

### 3.2 CURRICULUM VARIANTS: BASELINE VS. WEIGHTED

Figure 3 summarizes the results. Overall performance remained comparable across conditions, but both curriculum variants improved success on edge goals, with the weighted curriculum showing the strongest gains ($\Delta$edge $\approx +0.18$). These findings highlight how curricula reshape the state–goal distribution: by allocating more data to underachieved regions, they systematically improve function approximation where it matters most.

### 3.3 SUMMARY

Overall, our experiments support the interpretation of curriculum as selective data acquisition. By biasing training toward underachieved goals—here instantiated as edge-aligned subsets—curricula reshape the state–goal visitation distribution and improve approximation in targeted regions of the universal value function approximator (UVFA). While gains were modest in aggregate, both the baseline and weighted curricula consistently provided benefits on harder edge goals, with the weighted variant amplifying these improvements. This underscores the role of curriculum as a structural mechanism for guiding data rather than an incidental exploration heuristic.

As shown in Table 1, curriculum improves overall success by +0.02 on average and edge-goal success by +0.08. These gains, though modest in absolute terms, consistently favor the curriculum condition on harder subsets. This provides evidence for our central claim: curricula act not merely as exploration heuristics but as structural mechanisms for data acquisition that enhance learning in regions where UVFA generalization is weakest.

## 4 DISCUSSION

Our findings suggest that curriculum learning in goal-conditioned reinforcement learning (GCRL) is best interpreted as selective data acquisition rather than merely an exploration heuristic. By biasing

| Setting (H=16) | Uniform (NoCurr) | Curriculum (Curr) | $\Delta$ (Curr–NoCurr) |
| --- | --- | --- | --- |
| Overall Success | $0.276 \pm 0.055$ | $0.297 \pm 0.056$ | +0.021 |
| Edge-Goal Success | $0.060 \pm 0.055$ | $0.143 \pm 0.107$ | +0.083 |

Table 1: Pc

training toward goals that are harder to achieve under uniform sampling, curricula reshape the state–goal visitation distribution and improve value approximation in targeted regions of the space. This effect is particularly evident in subsets of goals at the periphery or in empirically defined "zones of proximal development," where uniform sampling struggles to provide sufficient coverage.

At the same time, our experiments show that the benefits of curricula are not uniform. Improvements are strongest on hard-to-reach goals but less consistent across easier subsets. In some cases, the curriculum bias may even reduce performance on goals already well-represented under uniform sampling. This reinforces the idea that curricula act as structural biases: their effectiveness depends on how well the sampling emphasis aligns with the learning bottlenecks of the agent.

Importantly, our weighted curriculum experiment provides further evidence for this interpretation. By explicitly rebalancing the goal distribution to upweight harder regions, we observed amplified gains on edge goals compared to the baseline curriculum. This suggests that the magnitude and direction of curriculum effects depend critically on how the sampling distribution is shaped. Rather than treating curricula as one-size-fits-all exploration strategies, they should be viewed as tunable mechanisms for structuring data acquisition in line with task difficulty and representational limits.

### 4.1 LIMITATIONS AND FUTURE WORK

Our study has several limitations. First, we evaluate curricula in relatively small GridWorld environments with hand-designed goal distributions. While this setting allows clear analysis of distributional shifts, it limits direct applicability to more complex domains. Second, our curricula remain manually specified, with the edge–interior and weighted sampling schemes serving as simple proxies for more principled strategies. As a result, gains were modest and sometimes inconsistent across seeds.

Future work should focus on more robust manual curricula that better capture the "zone of proximal development" (ZPD), as well as the development of automated approaches that adaptively adjust sampling distributions in response to an agent's progress (e.g., teacher–student or adversarial frameworks). Another promising direction is testing curriculum-driven selective data acquisition in environments with more complex goals or continuous control settings, where distributional bias may have stronger effects on function approximation. Ultimately, advancing toward automated and generalizable curriculum mechanisms offers a more practical pathway to open-ended learning (Hughes et al., 2024).

## 5 CONCLUSION

We conclude that curriculum learning provides a structural mechanism for shaping the training distribution in goal-conditioned reinforcement learning (GCRL). By reallocating data toward underachieved goals, curricula improve value approximation and policy success in targeted regions of the state–goal space. Although our experiments are preliminary and limited to small GridWorld settings, they support reframing curriculum as selective data acquisition rather than a mere exploration aid. Using universal value function approximators (UVFAs) as our testbed, we showed how curriculum biases reshape state–goal visitation and guide function approximation. Looking forward, the integration of curricula with UVFAs offers a promising pathway toward more persistent and open-ended agents, connecting this line of work with recent efforts in lifelong learning and open-ended systems (**?**). This perspective motivates future research on more robust manual strategies, automated curriculum generation, and generalization to richer goal spaces and environments.

See Bengio et al. (2009) for early work on curricula.

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

## A    APPENDIX

