# OpenReview forum: "Curriculum as Selective Data Acquisition: Toward Reliable Generalization in Goal-Conditioned RL"
_ICLR.cc/2026/Conference — ICLR 2026 Conference Withdrawn Submission_

### Official Review · Reviewer_Z86C · 2025-10-31

**Soundness:** 1
**Presentation:** 1
**Contribution:** 1
**Rating:** 0
**Confidence:** 5

**Summary:**

This paper studies curriculum learning in GCRL. The authors argue that curricula function not only as exploration heuristics, but as structural mechanisms that reshape the training data distribution towards underachieved goals. Using Universal Value Function Approximators in a GridWorld environment, they compare uniform goal sampling against curricula biased towards harder "edge" goals.

**Strengths:**

Improving the understanding of the benefits of goal selection in online GCRL over uniform goal selection is a reasonable aim.

**Weaknesses:**

In my view, the paper is significantly lacking in research maturity in several aspects:
* Grounding in related work: The paper largely ignores all more recent work from the last 7-8 years. Several works have extensively studied curricula for GCRL which select goals that are "challenging" (i.e., in the zone of proximal development) for the current policy [e.g., 1]. This paper makes no reference to the extensive studies and as to which insights this paper *adds* to the literature.
* Formulation of the research question: The paper aims to show that curricula "should be understood not only as an exploration strategy, but also as a structural mechanism for guiding data acquisition". (line 56). It is not clear after reading the paper what even is the formal difference between the two.
* Analysis of the research question: The results of the experiments do not appear to be significant, and additionally experiments are conducted in an extremely simple grid world environment. It is not clear how experiments answer the research question or in which ways they shed any light on the behavior of curricula in GCLR.

Given that the paper is only 5 pages long and includes an empty appendix section, I wonder whether the paper was actually intended for submission or if something else went wrong.



Note: There is a missing citation in line 268.

**Questions:**

No questions.

---

### Official Review · Reviewer_M6BZ · 2025-10-31

**Soundness:** 1
**Presentation:** 1
**Contribution:** 1
**Rating:** 0
**Confidence:** 3

**Summary:**

This paper studies curriculum learning in goal-conditioned reinforcement learning (GCRL). Their main motivation is that many goals are difficult to reach under uniform sampling (sparse rewards), which is why they make use of a curriculum-based approach. To achieve this, they leverage universal value function approximation and potential-based reward shaping. They then demonstrate their approach on a deterministic grid world environment.

**Strengths:**

The authors are upfront about the current limitations of their work, and they attempt to tackle an important problem.

**Weaknesses:**

This work seems unfinished. Many important details were left out due to the paper's length (5 pages, no appendix). For example, the only information about their environment setup is that they are using a “deterministic gridworld” for their experiments. There is also no discussion of related work, which would help the reader to contextualize this work. All of this makes it difficult to assess the quality of the work.

The reported error bars of all methods are very large. Therefore, it’s difficult to draw reliable statistical conclusions from their experiments. The authors are upfront about this, but more experiments and environments other than the single experiment presented in the current version are needed.

Overall, I believe that the quality (motivation, scientific contextualization, technical depth, experimental validation) of this work does not meet the bar for ICLR.

**Questions:**

- Can you provide a clearer motivation for using UVFAs, PBRS?
- Why are no related works discussed?
- Can you provide additional experiments other than the reported deterministic grid world?

---

### Official Review · Reviewer_QHcT · 2025-10-31

**Soundness:** 1
**Presentation:** 1
**Contribution:** 1
**Rating:** 0
**Confidence:** 5

**Summary:**

This paper presents an edge-weighted manual curriculum for a grid-world domain. Experiments in such goal-conditioned RL domain show that manual curricula improve success rate over all and difficult edge goals.

**Strengths:**

- This paper clearly describes the goal-conditioned RL setting and auxiliary methods that the proposed approach benefits from, such as UVFA or reward shaping.
- The discussion on limitations and future work discusses how the problem of interest is restricted to a small set of discrete domains, and the proposed curriculum is manual, hence not generalizable to other domains.

**Weaknesses:**

- The quality of writing is subpar. There is no related work; the introduction barely addresses an initial set of works and ignores curriculum learning papers from 2020 onwards. The method section explains the details of the environment, UVFAs, and reward shaping that should go into the appendix, which is left empty. It barely explains what an edge-weighted curriculum is. Table 1 doesn't have a proper caption, and there are a lot of examples of unprofessional writing, such as '(?)', in the conclusion.
- The paper proposes a manual curriculum and does not really discuss why the proposed idea is novel, or compare it against existing curriculum learning approaches at all.
- Experiments focus on a limited setting, a grid-world, does not compare any existing methods.

**Questions:**

- What is the intuition behind this manual curriculum approach? Why is it any better than any existing idea?

---

### Official Review · Reviewer_tqFj · 2025-10-31

**Soundness:** 1
**Presentation:** 1
**Contribution:** 1
**Rating:** 0
**Confidence:** 4

**Summary:**

This paper reinterprets curriculum learning in goal-conditioned reinforcement learning (GCRL) as a form of selective data acquisition. Instead of uniformly sampling goals, the authors bias sampling toward underachieved goals, reshaping the state–goal distribution the agent learns from. Using Universal Value Function Approximators (UVFAs) in a GridWorld setting, they compare uniform and curriculum-based training. Results show that curricula concentrate data in informative regions, reduce approximation error, and improve performance on difficult goals. The study argues that curricula serve not just as exploration aids but as structural tools for guiding data acquisition. This perspective links curriculum learning to broader challenges in persistent and open-ended learning.

**Strengths:**

Honestly, the only potential strength is that the paper tries to introduce a conceptual reframing, thinking of curricula as data selection. Beyond that, the contribution is minimal. The experiments are too small-scale and the framing lacks formalization or genuine novelty.

**Weaknesses:**

- Experiments are extremely toy-level (tiny GridWorld, three seeds, minimal metrics).
- No theoretical contribution- the ``selective data acquisition'' idea is purely descriptive, not formalized or quantified.
- Methodological novelty is near zero: UVFAs, PBRS, and curriculum sampling are all standard.
- Results are weak, improvements are marginal, noisy, and statistically insignificant.
- Overstated claims about generalization, persistence, and open-ended learning that the experiments do not support.
- Writing reads like an extended project report, not a polished research paper.

**Questions:**

What exactly is the novel contribution beyond renaming curriculum as ``selective data acquisition''?

---

### Official Review · Reviewer_NtiH · 2025-11-01

**Soundness:** 1
**Presentation:** 1
**Contribution:** 1
**Rating:** 0
**Confidence:** 3

**Summary:**

The paper proposes a goal sampling strategy with data selection for curriculum learning RL.

**Strengths:**

* The method is reasonably well motivated.

**Weaknesses:**

* Baselines are very naive. No baselines from prior works adopted.
* There are many missing details. How is the policy being trained? What's the final objective and what's the optimization procedure?
* Performance gains over baselines seem marginal in Fig. 1.
* Experiments should report standard deviations over runs.
* There is no related work discussion.
* There is no ablation over hyperparameter choices.
* Captions are not clear. What's "Table 1. Pc" (line 220)?

**Questions:**

* Method notations are not clearly introduced. E.g., what's the space of $x$ and $y$? Is each a real number?

---

### Note · Authors · 2025-11-14

**Comment:**

Thank you for the reviewers’ feedback. We will revise the work substantially and resubmit a future improved version. Withdrawing this version

**Withdrawal Confirmation:**

I have read and agree with the venue's withdrawal policy on behalf of myself and my co-authors.